# Optical Fiber Probe with Integrated Micro-Optical Filter for Raman and Surface-Enhanced Raman Scattering Sensing

**DOI:** 10.3390/nano14161345

**Published:** 2024-08-14

**Authors:** Md Abdullah Al Mamun, Tomas Katkus, Anita Mahadevan-Jansen, Saulius Juodkazis, Paul R. Stoddart

**Affiliations:** 1School of Science, Computing and Engineering Technologies, Swinburne University of Technology, John Street, Hawthorn, VIC 3122, Australia; mmamun@swin.edu.au; 2Optical Sciences Center, Swinburne University of Technology, John Street, Hawthorn, VIC 3122, Australiasaulius.juodkazis@gmail.com (S.J.); 3Biophotonics Center, Vanderbilt University, Nashville, TN 37235-1631, USA; anita.mahadevan-jansen@vanderbilt.edu

**Keywords:** Raman scattering, surface-enhanced Raman scattering, double-clad fiber, optical nanosensing, microfilter, femtosecond laser micromachining

## Abstract

Optical fiber Raman and surface-enhanced Raman scattering (SERS) probes hold great promise for in vivo biosensing and in situ monitoring of hostile environments. However, the silica Raman scattering background generated within the optical fiber increases in proportion to the length of the fiber, and it can swamp the signal from the target analyte. While filtering can be applied at the distal end of the fiber, the use of bulk optical elements has limited probe miniaturization to a diameter of 600 µm, which in turn limits the potential applications. To overcome this limitation, femtosecond laser micromachining was used to fabricate a prototype micro-optical filter, which was directly integrated on the tip of a 125 µm diameter double-clad fiber (DCF) probe. The outer surface of the microfilter was further modified with a nanostructured, SERS-active, plasmonic film that was used to demonstrate proof-of-concept performance with thiophenol as a test analyte. With further optimization of the associated spectroscopic system, this ultra-compact microprobe shows great promise for Raman and SERS optical fiber sensing.

## 1. Introduction

Interest in optical fiber probes for use in Raman and surface-enhanced Raman spectroscopy remains strong due to several practical advantages. For example, they are relatively non-invasive for in vivo applications, they provide sampling versatility, multiplexing, and distributed sensing capabilities, they may offer a power delivery advantage, and they allow for the collection of signals from highly scattering media [1,2]. They can also be extended for remote sensing and in situ monitoring of hostile environments [3]. In conventional Raman spectroscopy, the adjustment of the optical sampling system is usually tedious and requires specialist skills, whereas optical fibers greatly facilitate optical adjustment and spectral collection [4]. However, an issue with fiber optic Raman sampling is the background signal generated by Raman scattering within the fiber itself. Scattering that occurs within the numerical aperture (NA) of the fiber is efficiently returned to the spectrometer. Furthermore, the silica spectrum is broad and covers most of the useful Raman spectral range with varying intensity. While the strongest features occur below ∼1100 cm^−1^, weak scattering is present over the entire 0 to 3500 cm^−1^ range. This background signal is structured, making effective subtraction difficult and the detection of weak signals from the target analyte molecules is compromised or at times impossible [5]. The extent of interference from fiber scattering increases with fiber length, which also limits the maximum length of the fiber probe.

Several previous investigations have sought to minimize the effect of the silica Raman scattering background on the observed spectrum. Firstly, the fiber material can be selected to minimize the Raman background [4,6,7,8,9]. A study on the effect of silica Raman scattering on sample spectra for an unfiltered 18-around-1 probe found that if low hydroxyl silica fibers are used the background can be reduced [10]. Similar experiments to determine the most appropriate type of fiber for surface-enhanced Raman scattering (SERS) measurements were conducted using a 1-in/1-out probe. Bello and Vo-Dinh compared SERS spectra and the fiber Raman background obtained with silica-clad and polymer-clad silica fibers, concluding that the silica-clad fiber was the most efficient for both excitation and collection purposes [11]. Alternatively, the interference from the silica Raman scattering can be reduced by improving the collection efficiency of the analyte signal. In the single-fiber arrangement, it has been shown that the inner cladding of the double-clad fiber (DCF) can be used to increase the coupling efficiency of the Raman-scattered signal. It was found that DCF could yield up to a twelve-fold improvement in the signal-to-background ratio compared to conventional single-mode fiber and hollow-core photonic crystal fiber [12].

Komachi et al. [13] used a hollow waveguide, which could reduce the silica Raman background to some extent but had lower transmission efficiency than silica fiber. Furthermore, the waveguide was less flexible, which limits its applications in many practical sensing scenarios. Konorov et al. [14] used a hollow-core photonic crystal fiber for delivering the excitation laser and three standard silica-core fibers for collecting the Raman-scattered signal. More recently, the complexity of these multiple-fiber arrangements has been considerably reduced through the use of hollow-core negative-curvature fibers [15]. This type of fiber reduced the background Raman emission by at least 1000 times compared to conventional solid fiber. Subsequent work added a micro-lensed cap to improve the collection efficiency by 50% compared to a cleaved endcap [16] and packaged the 250 µm fiber into sub-millimeter tubing to make a Raman probe compatible with standard bronchoscopes [17].

Another major approach to the reduction of the background is to place filters at the distal tip of the Raman probe. For this purpose, two different filters are used at the end of a probe. One is a bandpass or short-pass filter (SPF) that serves to block the fiber Raman background generated by forward scattering in the excitation/delivery path. The other is a notch or long-pass filter (LPF) that excludes reflected Rayleigh-scattered light to prevent the generation of the fiber Raman background in the collection path. However, there are two major problems with filtered probes. Firstly, the need for external filters at the ends of the excitation and collection fibers adds extra bulk to the fiber probe. Secondly, for efficient performance of external-filtered probes, there is a trade-off between the collection efficiency of the focusing lens and the need for collimated light to deliver effective filter performance [18]. Myrick and Angel provided an early demonstration of filtering methods for reducing the silica Raman background in optical fiber probes [5]. Their designs relied on filtering the output of the excitation and the input of the collection fibers. Carrabba and Rauh reported a coaxial Raman fiber probe using discrete lens and filter technology similar to that of Myrick and Angel, in which the two arms of the probe were integrated so that the excitation and collection radiation shared an identical beam path over some of the probe’s length [19]. Several attempts have been made to fabricate an optical fiber probe with in-the-tip filters for suppressing the Raman background generated in the optical fibers [7,20,21]. These generally depend on a single fiber to deliver the excitation laser light, while multiple fibers are used to collect Raman-scattered signals from the sample. To our knowledge, the design of Komachi et al. [7] has achieved the smallest reported diameter of 600 µm for a filtered optical fiber Raman probe, based on eight collection fibers arranged around a central delivery fiber. Background reduction was achieved through a concentric arrangement of precisely machined bandpass (center) and long-pass (circumferential) filters butted up directly against the fiber end faces.

In this work, to further reduce the probe diameter from the record set by Komachi et al. [7], a novel double-sided filter has been designed and fabricated for positioning in-line with a DCF tip. To achieve a double-sided filter, a SPF coating was deposited onto the polished back surface of a LPF substrate. Femtosecond laser machining was then used to pattern the filters in a coaxial arrangement so that the SPF alone transmitted the excitation beam from the DCF core to the sample, and the LPF prevented the Rayleigh scattering from entering the inner cladding in the collection path. After aligning and attaching the filter assembly to the fiber tip, the performance of this assembly was evaluated by means of preliminary SERS measurements with thiophenol as a test analyte.

## 2. Fabrication of the Microfilter Probe

The working principle of the double-sided microfilter assembly is illustrated in Figure 1. Moving from left to right, after the double-clad fiber (DCF), a first glass plate supports an SPF on the first surface and an LPF on the second surface. A ring of the SPF coating is removed, leaving a central island of filter that matches the diameter of the DCF core. Additionally, a hole with a similar diameter to the short-pass island is drilled into the LPF coating, with the hole centers coaxially aligned. After fabrication, when the short-pass side of the microfilter assembly is attached to the DCF tip, the island of SPF passes only the excitation laser light exiting from the core while blocking the accompanying silica Raman background signal at higher wavelengths.

Upon emerging from the single-mode core, which is more accurately described as “few mode” at the wavelengths used here, the excitation beam will expand in the index-matched silica plates. Assuming a Gaussian beam profile, the spreading of the beam due to diffraction ($\theta_{d}$) can be approximated by [22]
(1)$$\theta_{d}≅\frac{\lambda }{\pi na},$$
where *λ* is the wavelength of the light, n is the refractive index of the glass plate, and a represents the beam waist radius (approximated here as the DCF core radius). Due to practical limitations in reducing the thickness of the filter plate to less than about 0.7 mm (see Section 2.1), Equation (1) suggests that the beam radius would expand to approximately 40 µm over that distance. Machining a hole of that size into the LPF would reduce the effectiveness of the LPF in blocking reflected excitation light from returning into the inner cladding. Therefore, a decision was made to restrict the diameter of the hole to a range of 10–20 µm.

After passing through the hole in the LPF, the excitation beam continues through a second UV-grade fused silica plate, which is sufficiently thick so that the beam is able to expand, and the resulting scattered light fills the collection aperture of the inner cladding of the DCF. The thickness of this spacer plate is somewhat dependent on the transparency of the sample, as discussed in [23,24]. The sample and the SERS substrate (if present) can be placed at the free surface of the second glass plate. Tracing the Raman or SERS signal back towards the inner cladding of the DCF, the bulk of the LPF is still present to remove reflected and scattered light at the excitation wavelength returning from the sample/SERS substrate. Reflected and Rayleigh-scattered light from the sample has a much higher intensity than the Raman-scattered signal and, if neglected, can generate a significant proportion of the silica background in a single-ended optical fiber Raman probe. The scattered light coupled into the inner cladding of the DCF is then fed back to a spectrometer via a double-clad fiber coupler. The steps that were taken to fabricate the double-sided filter are shown in Figure 2 and described in more detail in the following sections.

In principle, using appropriate masks, it may be possible to achieve a single-sided filter by depositing the LPF directly into the ablated (or etched) ring of the SPF. However, in our preliminary testing, we found that the shadowing effect at the filter edges during coating deposition extends over larger distances than the size of the target structures. Indeed, where lithographic patterning of filters was previously applied to a Raman probe, the target structures were significantly larger than the discrete 125 µm diameter fibers used in a relatively bulky seven-around-one design [25]. Therefore, we have focused on the double-sided design described here.

### 2.1. Preparation of Filter Substrate

A commercially available LPF suitable for Raman spectroscopy was used as the starting point for this proof-of-concept study (Edmund Optics, Stock No. 84-744, OD ≥ 4, cut-on wavelength 525 nm, 25 mm diameter). The rear surface of this filter was used for the SPF deposition, as described below. The initial thickness of the UV-grade fused silica substrate was 3 mm. To ensure efficient light collection by the inner cladding of the DCF, the substrate thickness was reduced through grinding and polishing in order to reduce the divergence of the light passing through it. On the other hand, the substrate thickness had to be at least 0.7 mm to avoid warping due to differential thermal expansion during coating [26].

To grind and polish the glass substrate, the filter was mounted with dental modeling wax on a block of 25 mm diameter. The wax layer also served to protect the LPF coating during polishing. Metalog Method G [27] was used as an initial guide for grinding and polishing, as summarized in Table 1. Grinding and polishing with a test substrate suggested that an additional 3 μm fine grinding step was required between the 9 and 0.04 μm steps. Once the polishing was completed, the block was heated to melt the wax, and the substrate was detached from the mounting block. The polished surface was then used for depositing the SPF coating (Figure 2).

**Figure 2 nanomaterials-14-01345-f002:**
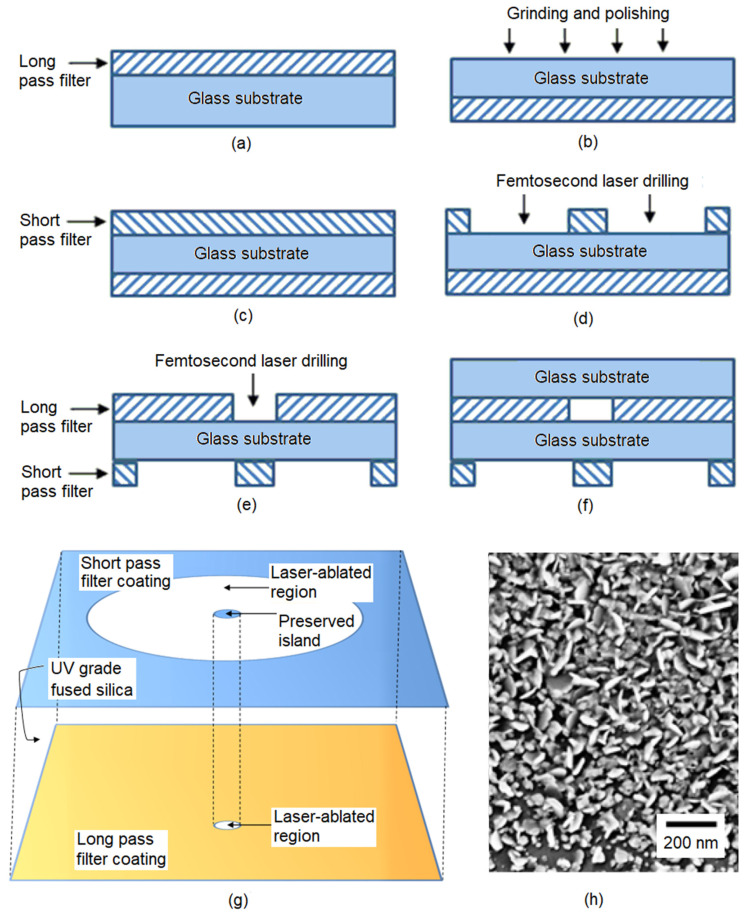
Schematic illustration of the sequence of fabrication steps used to form the double-sided microfilter assembly for use on a DCF fiber tip (figure not to scale). (**a**) A commercially available LPF was used as the starting point (see text for details). (**b**) The glass substrate was ground down and polished to reduce the thickness of the substrate to approximately 0.7 mm. (**c**) The SPF was deposited onto the opposing surface to the long-pass coating. (**d**) A ring of SPF was removed through femtosecond laser drilling, after which (**e**) the hole in the LPF was drilled as described in the text. (**f**) Finally, a further glass spacer was bonded to the LPF surface to provide a substrate for the SERS sensing surface. (**g**) Perspective view of the filter patterns. (**h**) Scanning electron microscope image of SERS-active, photochemically deposited silver nanoparticles on the surface of the outer glass substrate.

The 28-layer SPF was designed and fabricated by Optical Coating Associates (Queensland, Australia) with a cut-off wavelength of 516 nm. Note that the performance in terms of edge steepness and optical density of both filters could be substantially improved through the selection and fabrication of more sophisticated designs from specialist suppliers. However, the specifications selected here were deemed appropriate for the exploratory nature of the study.

### 2.2. Micro-Patterning of Filter Coatings

Femtosecond laser ablation was used to pattern the filter coatings [28] using a PHAROS 230 fs laser (Light Conversion, Vilnius, Lithuania) operating at 600 kHz with a 5× objective lens (0.14 NA, Mitutoyo, Kanagawa, Japan). The focused light spot diameter was calculated to be approximately 4.5 μm. The desired machining sequence was drawn and programmed in the laser system control and automation (SCA) software (WOP, Vilnius, Lithuania). Laser machining was performed on both sides of the filter, according to the schematic given in Figure 3. First, the SPF coating was ablated in a ring with an inner diameter of approximately 16 μm and an outer diameter of ∼110 μm. Then, the sample was flipped over, and the focal point was aligned with the center of the ring pattern as a fiducial mark to realign the X and Y coordinates. The focal plane was then shifted by changing only the Z coordinate for drilling a 16 μm diameter hole into the LPF coating. Although the orthogonality of the XYZ stage is high (nominally 10 arc seconds or 0.003°), substrate tilt is likely to be much greater, and it is unavoidable. Conservatively estimating a substrate tilt of 0.1°, which is exaggerated compared to practical observations of the substrate surface in the focal plane, we would expect approximately 3.5 µm misalignment of the fabrication on each side of a 2 mm substrate. This can be compared with the calculated diffraction-limited laser beam waist diameter of 4.5 µm, and it is comparable to or less than the uncertainty of manual pattern alignment on the microscope camera. Importantly, any pattern misalignment can in principle be canceled out in the final assembly during the active fiber alignment process (see Section 2.3).

Pulse energies and patterning approaches were different for SPF and LPF processing. The initial fabrication on the SPF used 4.8 nJ pulse energy and a 1030 nm wavelength with concentric circular laser paths, whereas the LPF fabrication used 10.7 nJ pulse energy with a second harmonic 515 nm light and internally hashed circle patterns. Multiple ablation passes were used to achieve complete filter material removal without causing laser damage to the substrate (nine passes for the SPF and five passes for the LPF). The LPF hashing angle was changed with each pass to randomize the laser pulse placement. Other fabrication parameters were kept the same for both SPF and LPF fabrications; namely, six pulse bursts, 1.5 μm burst spacing along the laser paths, and 1.5 μm circle and hash line spacing. This spacing was selected to ensure a smoothly ablated surface based on sufficient overlap between ablation tracks for the given laser spot size. The laser wavelengths and powers used for drilling the two filters are listed in Table 2.

One of the by-products of femtosecond laser machining is the deposition of debris on nearby surfaces. This debris was removed through a two-step cleaning process. Firstly, the filter was sonicated in acetone and isopropyl alcohol, each for 10 min. It was then rinsed with Milli-Q water three times. In the second step, the substrate surfaces were cleaned using Pelco cellulose acetate replicating tape (Ted Pella Inc., Redding, CA, USA). Before applying the tape, 10 μL of acetone was pipetted onto the substrate surface. Then, a section of the tape (30 mm × 30 mm) was carefully applied to the substrate surface from one end to the other so that no air was trapped underneath. The tape was then allowed to dry for 15 min. When the dried tape was peeled off of the substrate surface, debris was removed with the tape to leave a cleaner surface. The cleaning process with the replicating tape was repeated three times to achieve a fully clean surface free of debris.

### 2.3. Fiber Mounting of the Microfilter Assembly

After patterning and cleaning the double-sided filter, UV-cured optical adhesive (Norland Optical Adhesive 61) was used to position a fused silica substrate of 1 mm thickness over the long-pass coating. This additional spacer layer provided a substrate for silver nanoparticle deposition and also allowed the Raman-scattered signal to completely fill the entrance aperture of the DCF inner cladding at the specified acceptance angle. The filter assembly was then diced into 2 mm × 2 mm squares (Disco DAD-321, Disco, San Jose, CA, USA).

To mount the microfilter assembly on the DCF (taken from a DC1060LEB coupler, ThorLabs, Newton, NJ, USA: core diameter 4 µm and NA = 0.19, inner cladding diameter 102 µm, NA = 0.24, outer cladding diameter 125 µm), a freshly cleaved tip was mounted in a fiber chuck on a five-axis positioning stage, comprising XYZ, tilt, and yaw movements. A diced microfilter cube was temporarily bonded with dental modeling wax to a glass slide that was in turn mounted on a second XYZ translation stage that provided a wider total adjustment range, as shown in Figure 4. The 514.5 nm beam from the argon-ion laser of a Renishaw in-Via Raman spectrometer was coupled to the free end of the DCF to assist with alignment. First, the DCF tip was brought close to the filtering assembly, which was then moved in the transverse plane to align the emission from the DCF core with the island in the short-pass coating and the corresponding hole in the LPF. Accurate manual alignment could be achieved by viewing the transmitted beam on a screen.

UV-cured optical adhesive (Norland Optical Adhesive 61) was now dispensed onto the DCF tip and the neighboring surface of the microfilter assembly. The glue was allowed to wick into the gap through small movements of the longitudinal translation stage axis. After visually confirming the alignment between the DCF core and the filter assembly via the transmitted laser beam spot, the UV glue was cured (ThorLabs CS2010 UV curing LED, ThorLabs, Newton, NJ, USA, 2 min exposure). The filter assembly was then detached from the glass slide by melting the wax bond with a heat gun. The integrated DCF–microfilter probe was then ready for deposition of the SERS substrate on the exposed silica surface.

### 2.4. Surface-Enhanced Raman Scattering

The silver nanoparticle (AgNP) SERS substrate was deposited onto the exposed silica surface by means of the photochemical deposition technique described in [29]. Briefly, the distal tip of the DCF+microfilter assembly (or DCF+blank for control samples) was placed in an aqueous growth solution of equimolar (1 mM) silver nitrate (AgNO_3_, >99.8%, Sigma–Aldrich, Castle Hill, Australia) and trisodium citrate (Na_3_C_6_H_5_O_7_, >99.0%, Sigma–Aldrich). The laser-induced deposition of AgNPs on the outer surface of the assembly was carried out using the 514.5 nm emission of an Ar+ laser focused into the DCF core at the proximal end. For convenience, we used the laser in the In-Via Raman microscope (Renishaw plc, Wotton-under-Edge, UK). Photo-induced AgNP deposition occurred in the circular illuminated region on the glass surface during a 4 min irradiation period, as shown in Figure 2h. The particle size ranged from 20 to 100 nm with a distinct peak in the distribution at 85 ± 5 nm. This technique has been shown to generate heterogeneous aggregates and intergrowths that scatter light over a broad range of excitation wavelengths as a result of a wide variety of possible surface plasmon resonances [30]. Following deposition, the fiber segment was taken out and carefully rinsed with deionized water to remove excess silver ions and citrate from the deposited film and then blow-dried with pure N_2_.

Raman spectra were acquired with the same in-Via Raman microscope using an excitation wavelength of 514.5 nm, 0.6 mW power, 10 s acquisition time, and one accumulation. The beam was focused on the DCF core via a 10× objective lens (NA 0.25), which was also used to collect the backscattered Raman signal in this simplified setup. A 2400 groove per millimeter grating dispersed the Raman spectra onto the CCD detector, with the spectrometer entrance slit set to standard confocal mode (65 μm). This setup preferentially samples the signal coupled back through the DCF core, but it was deemed adequate for this initial evaluation (see Section 3 for further discussion). Thiophenol was used as a test analyte for SERS characterization as it provides strong characteristic peaks and forms a consistent self-assembled monolayer on silver surfaces [31]. A 10 mM ethanolic solution was prepared from thiophenol (99%+, Sigma–Aldrich). All solvents were 99% or higher purity and used without further treatment.

## 3. Results and Discussion

The transmission characteristics of both filter coatings were evaluated in a Cary 50 UV-Vis spectrophotometer (Agilent, Melbourne, Australia). This was performed to confirm the filter performance and to find the wavelength of highest absorption suitable for femtosecond laser ablation. During deposition of the SPF, a test window was included as a witness plate. As shown in Figure 5, there was a narrow peak in the transmission spectrum of the short-pass test window around the intended excitation wavelength (514.5 nm) and low transmission in the target Raman spectral range above 525 nm (Raman shifts ≥ 400 cm^−1^). This confirmed the success of the filter-coating process. The relatively simple 28-layer filter design used here (see Section 2.1) was unable to provide uniformly high transmission throughout the short wavelength pass band, but it successfully reduced the risk of coating errors and was deemed acceptable for this exploratory study. In contrast, the LPF coating passes wavelengths above 525 nm and blocks all transmission below 525 nm. In the double-sided filter, the transmission characteristics of both short- and long-pass coatings combine to effectively block much of the visible spectrum, which confirmed the performance of the combined short-pass and long-pass coatings.

An optical profilometer (Veeco Contour GT-K, Bruker, Billerica, MA, USA) image of a typical short-pass island and ablated region is given in Figure 6a,b. It is seen that the outer diameter of the ablated ring is around 110 μm, while the island diameter in this example was set to 16 μm. The depth of the ablated region is ∼3.7 μm, which matches the thickness of the short-pass coating measured at the masked edges. Although we were unable to directly measure the transmission spectrum of the ablated regions, epi-illumination with white light resulted in a blue appearance (Figure 6c), which is consistent with wavelengths above ~520 cm^−1^ being transmitted by the LPF on the other side of the plate while the shorter blue–green wavelengths are reflected. This image also shows faint traces of the circumferential tool path of Figure 3a. Further detail of this pattern can be seen in the scanning electron microscope (SEM; Raith 150 Two, Raith, Dortmund, Germany) secondary electron image in Figure 6d. Here, it appears that each laser pulse in the final pass leaves a mostly clear spot surrounded by some small 50–100 nm debris particles on the glass surface. Due to weak interactions with light, these small sub-wavelength debris particles cannot be removed by the laser, nor are they dislodged by the cleaning protocol described in Section 2.2, which was successful in removing the larger debris particles from the short-pass island surface, the ablated region, and the surrounding short-pass coating surfaces. There is no evidence of any residual layers of the SPF left in the ablated region, whereas the layered structure of the filter coating is clearly visible in the edges of the central island. The edges of the island extend over a distance of about 2 µm.

To quantify the performance of the DCF probe with the double-sided microfilter assembly, it was necessary to compare it with a suitable control. Therefore, a fused silica substrate with the same total thickness as the microfilter assembly was used as a control substrate attached to a second length of DCF. AgNPs were then deposited onto both microfilter assembly and control substrates using the procedure described in Section 2.4, and the resulting SERS spectra of thiophenol were recorded. Three different lengths (12, 20, and 25 cm) of DCF were evaluated. The characteristic thiophenol peaks were visible in all samples, with the spectrum obtained from the longest 25 cm DCF microprobe shown in Figure 7a. The characteristic thiophenol peaks are clearly visible in Figure 7b after subtraction of the fiber Raman background, but they are not visible in any of the DCF controls with silica blanks. The peaks shown here correspond to ring out-of-plane deformation and C–H out-of-plane bending (997 cm^−1^), ring in-plane deformation, C–C symmetric stretching (1021 cm^−1^), C–C asymmetric stretching (1071 cm^−1^), and C–C symmetric stretching (1572 cm^−1^) [32,33].

The spectrum shown in Figure 7a was obtained under a Raman microscope, where the excitation is coupled directly into the DCF core and the returned Raman signal is collected with the same objective. Although we have used the relatively relaxed “standard confocality” setting of the in-Via spectrometer (slit width = 65 µm), this arrangement favors signal collection back through the core—rather than the inner cladding—and thus maximizes the background collection at the expense of the SERS signal. This interpretation is confirmed by Figure 7c, where the fiber Raman background intensity is seen to scale approximately linearly with the length of the probe, while the SERS intensity is relatively constant with length. Here, the background has been quantified from the height of the broad silica peak at about 810 cm^−1^, while the SERS intensity has been inferred from the average of the peaks at 997 and 1572 cm^−1^. In both cases, the peaks’ intensities have been normalized against the 12 cm probe length to assist with comparison. While this simplified setup was used here for proof of concept, in practice, this kind of probe would be deployed through a 2 × 1 optical fiber coupler that would (i) separate the excitation and collection paths, thus reducing the amount of fiber Raman background coupled back to the detector, and (ii) allow the full aperture of the inner cladding to be coupled into a spectrometer that is optimized for this kind of extended source.

The thiophenol spectrum could not be resolved through control probes that were terminated with a blank silica plate instead of the microfilter assembly. This is presumably due to the additional background generated by Rayleigh-scattered laser light on the return path drowning out the relatively weak SERS signal in this unoptimized setup. Therefore, to further quantify the performance of the microfilter, we have compared the signal-to-background ratio of three 12 cm DCF microfilter probe samples with previously published results obtained from an unterminated DCF segment of 2.5 cm length [10]. For consistency in calculating the signal-to-background ratio, the signal was taken as the height above baseline of the thiophenol SERS peak at 1572 cm^−1^, and background as the corresponding measure of the broad silica Raman peak around 810 cm^−1^. This measure was specific to the test analyte thiophenol used here, but it allows a consistent comparison between the probes with microfilters and the bare DCF probe (without microfilters) reported in [12]. The peak intensities were normalized against the excitation power and spectral accumulation time, while the background peak intensity was also normalized against the length of the probe. The normalization for probe length is justified by the background scaling linearly with length, as shown in Figure 7c. Baseline subtraction was performed by the penalized least squares method with adaptive weighting described in [34]. The spectra were smoothed by using a Savitzky–Golay filter (third-order polynomial, seven-point window). The results obtained from the calculation of signal-to-background ratio in the 12 cm DCF probe with microfilter assembly are given in Table 3.

With the background intensity normalized for probe length, the average signal-to-background ratio obtained from three different DCF microfilter probes of 12 cm was 7.1 ± 2.3, while it was approximately 3.0 for the bare DCF probe reported in [12]. Thus the microfilter assembly has approximately doubled the performance of the probe, based on this measure of length-adjusted signal-to-background ratio. This improvement is consistent with an effective suppression of fiber Raman background generated by Rayleigh-scattered excitation light on the return path through the probe.

## 4. Conclusions

This work has shown that femtosecond laser micromachining can be used to accurately pattern filters for use in a miniaturized optical fiber Raman probe. Features as small as 10 µm in diameter were machined through laser ablation with edge transitions of approximately 2 µm. These dimensions are significantly smaller than those achieved either through conventional mechanical grinding [7] or lithographic techniques [Barton] and allow for coupling to a compact double-clad optical fiber probe. The resulting double-sided filter assembly could be accurately mounted on the DCF tip using a five-axis stage and visual feedback from the intensity of the transmitted laser beam. Subsequent SERS measurements from a self-assembled monolayer of thiophenol on a photochemically deposited silver nanoparticle substrate showed a two-fold improvement over the previously reported performance of an unfiltered DCF probe [12] based on the length-adjusted signal-to-background ratio. This is consistent with fiber Raman background excited by Rayleigh-scattered laser light being rejected by the filter on the return path through the probe, whereas the backscattered silica Raman generated in the excitation path is still collected by the microscope-coupled spectrometer arrangement used in this preliminary study.

These results suggest that the microfilter assembly has significant potential to improve the performance of a DCF-based Raman probe. Further improvements can potentially be achieved by connecting the probe via a 2 × 1 coupler to a single-mode laser source and a high-NA fiber-coupled spectrometer (see e.g., [20]). This arrangement, which will be the subject of future work, should serve to redirect background generated by the excitation in the DCF core away from the detector while ensuring that the signal is sampled from a greater extent of the inner cladding. Future work will also aim to assess the extent to which the performance of the filters is degraded for off-axis illumination, given the relatively large core and inner cladding of NAs. Previous reports, such as [7], suggest that this design can still achieve reasonably good filter performance due to the close proximity of the filters and the optical fiber.

As discussed in Section 2, the thickness of the spacer plate is somewhat dependent on the characteristics of the sample, including its transparency and refractive index. The overlap between separate delivery and collection fibers has been modeled in some detail for several cases, including conventional “six-around-1” geometries with both flat and beveled end faces [23,24]. Given the close analogy between an n-around-one design and the DCF geometry, it is reasonable to expect that beveling of the DCF end face would also improve the collection efficiency of the present probe in future work.

The results also indicate that the process of manufacturing the microfilters is reasonably cost-effective and scalable due to the number of filter elements that can be produced from each filter plate. Ongoing improvements in femtosecond laser technology are also promoting large area processing with high throughput [35]. The use of Bessel beams may allow for further reduction in the size of the damaged edge regions around the filter features [36]. Future work will explore these same laser-machining techniques for trimming the diced microfilters down to match the DCF outer diameter of 125 µm. At that scale, the probe would be compatible with standard 0.014” coronary guidewires, thus substantially reducing the size of current probes [7,17] and opening up new applications in diagnostics and process monitoring. Assembly of the DCF probe was also found to be repeatable, given the ability to align the DCF core and microfilter by means of the transmitted excitation. Importantly, the performance of the microfilter assembly can still be further improved by optimizing the SPF design and the physical dimensions of the structure, e.g., substrate thicknesses, the SPF island diameter, and the long-pass hole diameter. In typical guidewire applications, we expect the pathlength in the optical fiber to be approximately three orders of magnitude longer than that in the filter assembly. However, if the optical fiber Raman background can be sufficiently suppressed, then the scattering contribution from the filter substrate and spacer plate might become the dominant contribution. In that case, these optical elements could potentially be replaced by a material with intrinsically low Raman scattering, such as sapphire or Raman-grade calcium fluoride.

## 5. Patents

P.R. Stoddart, A. Mahadevan-Jansen, Md Mamun, and N.A. Cole, “Optical fibre based microprobe”, Australian Provisional Patent 25/01/2018. Assignee: Swinburne University of Technology.

## Figures and Tables

**Figure 1 nanomaterials-14-01345-f001:**
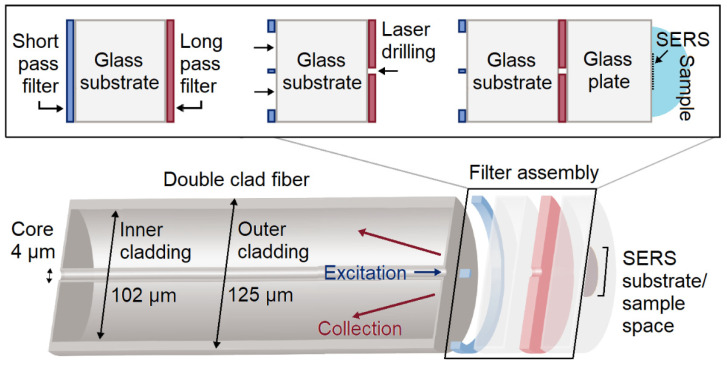
Schematic of the microfilter assembly on the DCF tip. Filter coatings are deposited onto both sides of a UV-grade fused silica substrate. A ring of the short-pass coating is ablated out to the diameter of the inner cladding, leaving an island at the center that blocks Raman-scattered light from the single/few-mode core. The LPF on the second side of the glass substrate has a hole drilled into the axis of the core to pass the clean laser excitation and reduce the intensity of Rayleigh-scattered light from the sample that returns to the inner cladding. Depending on the transmission characteristics of the sample, a second glass plate can be used to allow the Raman-scattered light from the sample to completely fill the aperture of the inner cladding. The SERS substrate is deposited onto the spacer plate as required.

**Figure 3 nanomaterials-14-01345-f003:**
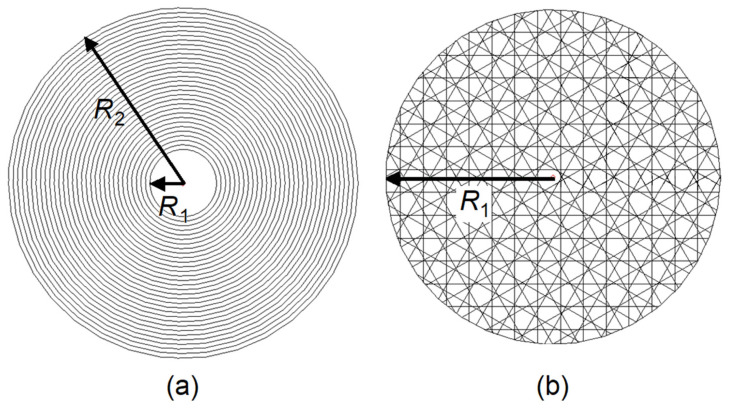
Tool paths programmed for ablating (**a**) the short-pass coating and (**b**) the long-pass coating. For the SERS testing presented below, *R*_1_ = 5 µm and *R*_2_ = 55 µm were used. The spacing and number of paths in each case were determined by the laser spot size (4.5 μm), the track overlap (1.5 μm), and the depth to be ablated.

**Figure 4 nanomaterials-14-01345-f004:**
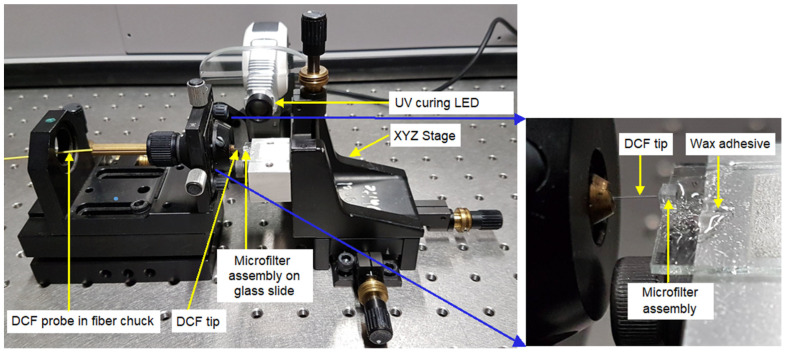
Translation stages and UV curing system used for aligning and attaching the microfilter assembly to the DCF tip.

**Figure 5 nanomaterials-14-01345-f005:**
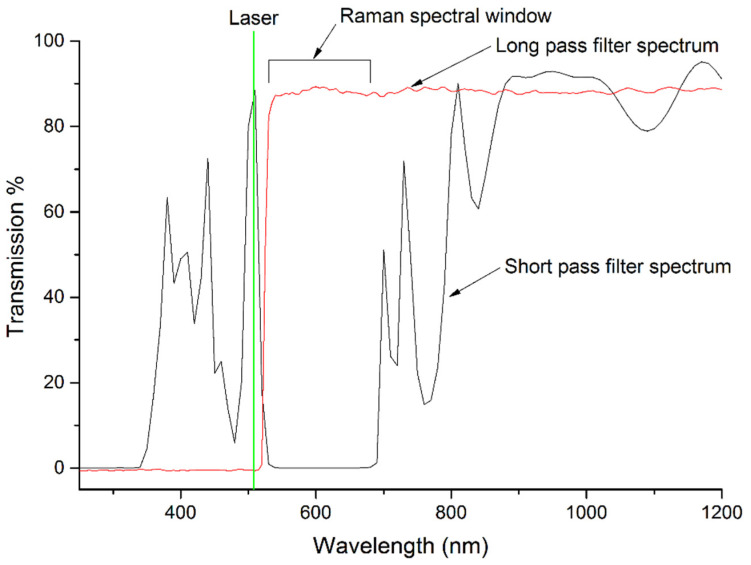
The double-sided filter combines the transmission characteristics of both the long- and short-pass filters. The SPF (black line) passes the laser line (shown in green) while blocking the silica Raman-scattered signal in the Raman spectral range. To enter the inner cladding of the DCF, Raman-scattered light from the sample passes through the LPF (red line) and through the ablated region of the short-pass coating, while Rayleigh scattering from the sample is blocked.

**Figure 6 nanomaterials-14-01345-f006:**
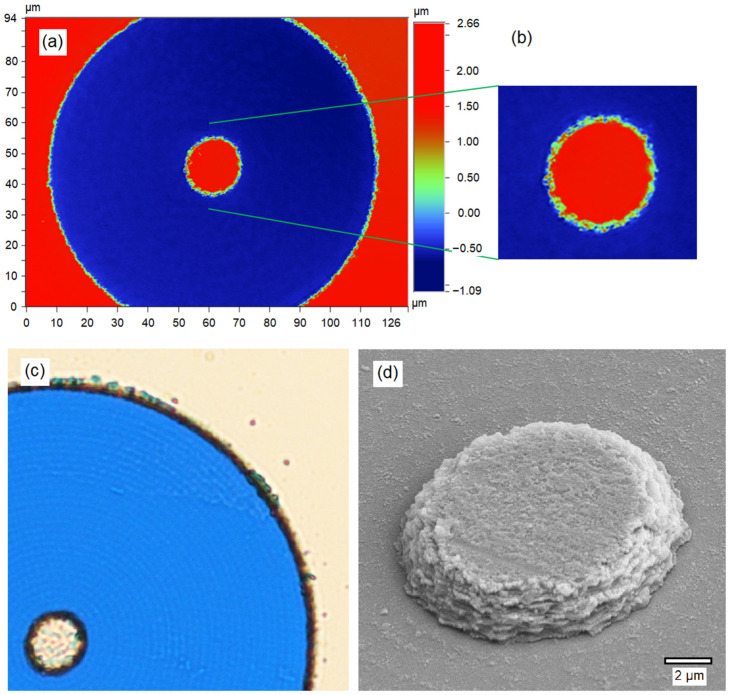
(**a**) Optical profilometer measurement of a typical short-pass island (*R*_1_ = 8 µm) and ablated ring after cleaning. (**b**) Magnified view of the short-pass island region from (**a**). (**c**) Microscopic image of another example with *R*_1_ = 5 µm, taken under white light epi-illumination (20× objective). Wavelengths above 520 nm are transmitted by the LPF on the far side of the plate, while the shorter, mainly blue wavelengths are reflected, resulting in the observed blue color of the ablated region. (**d**) SEM image of the short-pass island from (**c**), with the edges of the island showing some evidence of the discrete layers deposited to form the SPF. The ablated region is accurate to the design dimensions, and the boundaries between the ablated region and the remaining SPF are relatively narrow. While the circumferential tool path from Figure 3a can be discerned here in (**c**), and individual ablation sites can be seen on the glass surface in (**d**), there is no sign of any significant residual filter coating material on the ablated surface.

**Figure 7 nanomaterials-14-01345-f007:**
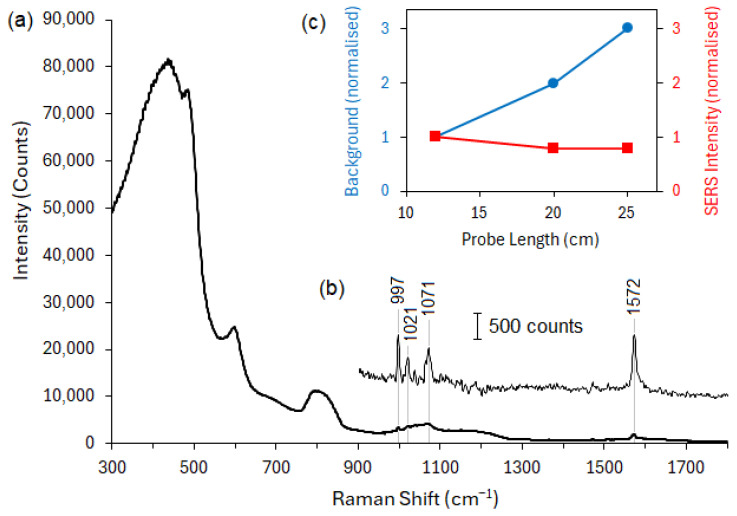
(**a**) Spectrum acquired through a 25 cm DCF segment with integrated microfilter assembly. (**b**) The characteristic SERS peaks of thiophenol are clearly visible after subtracting the fiber Raman background, which is generated primarily by the transmitted laser excitation in this simplified setup. The thiophenol spectrum could not be detected in any of the DCF probes without filtering assembly. (**c**) As expected, the intensity of the fiber Raman background scales approximately proportionally with the probe length, whereas the SERS peak intensity is reasonably constant with relatively minor losses for longer probe lengths. Peak intensities have been normalized against the 12 cm DCF probe in each case.

**Table 1 nanomaterials-14-01345-t001:** Metalog Method G for grinding and polishing [27] was modified through the addition of a 3 µm fine grinding step.

Step	Plane Grinding (PG)	Fine Grinding 1 (FG1)	Fine Grinding 2 (FG2)	Oxide Polishing (OP)
Surface	MD-Piano 220	MD-Plan	MD-Plan	MD-Chem
Abrasive	Diamond	DiaPro Plan 9	DiaPro Plan 3	OP-S Non Dry (colloidal silica)
Diamond grit/ grain size	68 µm	9 µm	3 µm	0.04 µm
Lubricant	Water	-	-	-
Rotational speed (rpm)	300	150	150	150
Force (N)/specimen	30	35	35	20
Time (min)	2:19	11:37	4:30	1:09

**Table 2 nanomaterials-14-01345-t002:** Typical laser parameters used for micro-patterning the filter coatings. For all drilling, the frequency was set at 200 KHz, and the laser was focused through a 5× objective with NA = 0.14.

Sample	Wavelength (nm)	Power (W)	Energy (nJ)	Attenuator (%)
Long-pass filter	515	2.5	10.7	40
Short-pass filter	1030	2.5	4.8	18

**Table 3 nanomaterials-14-01345-t003:** Signal-to-background ratio of the 12 cm DCF microprobes with attached microfilters, based on the thiophenol peak at 1573 cm^−1^ (signal) and the silica peak at 811 cm^−1^. The result from [12] for a bare DCF probe of 2.5 cm is reproduced here for reference.

Probe	Signal (Counts/mW/s)	Background (Counts/mW/s/cm)	SBR.cm
12 cm DCF microprobe	244 ± 35	34.5 ± 6.0	7.1 ± 2.3
2.5 cm DCF optrode [12]	20	6.8	3.0

## Data Availability

Raw data for the reported spectra are available from the corresponding author upon reasonable request.

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
