# Peer review of "Optical Fiber Probe with Integrated Micro-Optical Filter for Raman and Surface-Enhanced Raman Scattering Sensing"

_nanomaterials, 2024, doi:10.3390/nano14161345_

Round 1

Reviewer 1 Report

Comments and Suggestions for Authors

In this work, Al Mamun et al. developed and  fabricated a double-sided filter integrated on the tip of DCF probe via fs-pulsed laser micromachining. The filter was composed of a short pass and long pass microstructured coatings on both sides of a silica plate, which demonstrated good performance for SERS eliminating the inherent Raman scattering from the optical fiber. This work is interesting and well-organized. There are some comments to further improve the manuscript. 

1. As discussed in Line 140, the excitation laser continues through a fused silica plate as passing through the long pass filter side. Does the Raman scattering occur in the silica plate and interfere the Raman spectrum of the analyte?

2. The SEM micrographs of the structures on both sides of the filter should be provided to demonstrate the quality of laser micromachining. 

3. In Line 206, the authors claimed that the long pass filter coating was fabricated by shifting the focal plane down along the z axis. How did the author make the laser beam perpendicular to the silica plate surface? The effect of the tilting angle should be estimated.

4. In figure 5, the transmission of machined and unmachined coating for long and short pass filter should be plotted for comparison. In addition, why the short pass filter spectrum is not flat for the wavelength shorter than 520 nm.

Reviewer 2 Report

Comments and Suggestions for Authors

The article "Optical Fiber Probe with Integrated Micro-Optical Filter for Raman and Surface-Enhanced Raman Scattering Sensing" is dealt with development of the filter for the DCF fiber tip. The authors have studied and described in detail the technological process for producing such filter.  The obtained filter was tested in order to obtain the SERS signal of thiophenol sample. The spectral windows for long and short pass filter are obtained correctly. The paper is well-organized and well-written. In order to enhance the article it is recommended to add more information concerning Ag NP, interpretation of SERS peaks as well as to add more specific results in the conclusion section.

Please find below the full list of comments and suggestions.

1) line 14 here and further it is optionally recommended to replace Raman background with Raman scattering background

2) line 96 and further the abbreviation BP and especially LP should be used through the text (they were introduced at line 75 and 77)

 3) Figure 1 is needed in modification in SERS substrate part. In present version it is not so clear where the sample should be and where it is the Ag NP layer.

 It is recommended to show separately the sample layer, Ag NP layer as well as substrate at the SERS substrate.

 4) Figure 2 the authors should add the UV-Vis spectrum of AgNP in order to demonstrate the surface plasmon resonance (SPR) peak. It is necessary in order the readers can understand how the laser energy is correlated with the SPR peak absorbance (the laser excitation is close to the SPR absorbance peak maximum or far from it). Also the size as well as standard deviation or morphology description part about grown Ag NP is optionally recommended.

 5) the interpretation of thiophenol Raman peaks is needed at least in supporting materials.

 6) line 354-355 what is the relatively relaxed “standard  confocality” mean? What was the size (or diameter) of the confocal aperture?

7) line 386 Do the authors mean 7 point window in Savitzky–Golay filter? If so the window seven should be replaced with seven point window.

 8) The conclusion section should be rewritten.  It should reflect more specific results obtained in the study.

Round 2

Reviewer 2 Report

Comments and Suggestions for Authors

The authors responded to the comments and suggestions accordingly.

Line 418. Please check the reference (there are the surname of first author "[Barton]" instead of reference number).